# Characterization of Composition and Antifungal Properties of Leaf Secondary Metabolites from Thirteen Cultivars of *Chrysanthemum morifolium* Ramat

**DOI:** 10.3390/molecules24234202

**Published:** 2019-11-20

**Authors:** Huanhuan Xue, Yifan Jiang, Hongwei Zhao, Tobias G. Köllner, Sumei Chen, Fadi Chen, Feng Chen

**Affiliations:** 1College of Horticulture, Nanjing Agricultural University, Nanjing 210095, Chinachensm@njau.edu.cn (S.C.); chenfd@njau.edu.cn (F.C.); 2College of Plant Protection, Nanjing Agricultural University, Nanjing 210095, China; hzhao@njau.edu.cn; 3Department of Biochemistry, Max Planck Institute for Chemical Ecology, Hans-Knöll Str. 8, 07745 Jena, Germany; koellner@ice.mpg.de; 4Department of Plant Sciences, University of Tennessee, Knoxville, TN 37996, USA

**Keywords:** apolar secondary metabolites, terpenoids, antifungal, organic extract

## Abstract

*Chrysanthemum morifolium* Ramat is an ornamental plant of worldwide cultivation. Like many other species in the family Asteraceae, *C. morifolium* is a rich producer of secondary metabolites. There are two objectives in this study: (I) to determine and compare the diversity of apolar secondary metabolites among different cultivars of *C. morifolium* and (II) to compare their properties as antifungal agents. To attain these objectives, we selected 13 cultivars of *C. morifolium* that are commonly used for making chrysanthemum tea as experimental materials. Leaves at the same developmental stage were collected from respective mature plants and subjected to organic extraction. The extracts were analyzed using gas chromatography–mass spectrometry. A total of 37 apolar secondary metabolites including 26 terpenoids were detected from the 13 cultivars. These 13 cultivars can be largely divided into three chemotypes based on chemical principal components analysis. Next, the extracts from the 13 cultivars were examined in in vitro assays for their antifungal properties against three species of pathogenic fungi: *Fusarium oxysporum*, *Magnaporthe oryzae*, and *Verticillium dahliae*. Significant variability in antifungal activity of the leaf extracts among different cultivars was observed. The 13 cultivars can be divided into four groups based on their antifungal activities, which could be partly correlated to the contents of terpenoids. In short, this study reveals large variations in chemical composition, particularly of terpenoids, of leaf secondary metabolites among different cultivars of *C. morifolium* and their different abilities in functioning as antifungal agents.

## 1. Introduction

Plants make diverse phytochemicals with most of them defined as secondary, or specialized, metabolites [1]. Among the best-known classes of plant secondary metabolites are terpenoids, phenylpropanoids, and alkaloids [1,2]. Plant secondary metabolites have many biological and ecological functions, such as for defense against insect herbivores and pathogenic microorganisms and for establishment of mutualistic interactions [3,4,5,6,7]. In addition to their significance in fundamental plant biology and evolution, plant secondary metabolites have played an important role in human civilization. They have been widely used as medicines (e.g., artemisinin), spices (e.g., capsaicin), and agrochemicals (e.g., pyrethrin) [8,9,10]. Despite the wide applications of plant secondary metabolites, the vast diversity, biological functions, and potential applications of secondary metabolites from many plants remain poorly characterized. *Chrysanthemum morifolium* Ramat is one of them.

*Chrysanthemum morifolium*, also known as florist’s daisy, belongs to the Asteraceae family, which is arguably the largest family in eudicots [11]. Originated in China, several thousands of cultivars of *C. morifolium* have been developed. Most of the cultivars are used as garden plants or cut flowers. Nevertheless, some cultivars of *C. morifolium* have been developed for culinary and medical uses due to antioxidant and antimicrobial activities [12,13], especially as herbal teas made from flower heads [14]. In fact, the medicinal use of *C. morifolium* was noted in the first major work dedicated to Chinese Materia Medica known as Shennong Bencao Jing. Secondary metabolites from some *C. morifolium* cultivars have been analyzed. Pertaining to this study, essential oils extracted from some *C. morifolium* cultivars were shown to be dominated by terpenoids [15,16,17,18,19]. Nevertheless, systematic comparisons of composition of secondary metabolites and their biological roles from comparable cultivars are generally lacking.

We have started to systematically characterize the chemical composition, biosynthesis, and biological functions of apolar secondary metabolites in *C. morifolium*. The aim of the present study was to determine the chemical composition of apolar secondary metabolites from leaves of different cultivars of *C. morifolium* and to determine whether these compounds have any inhibitory effect on fungal pathogens. Such knowledge may facilitate the breeding of novel cultivars of *C. morifolium* with enhanced resistance to fungus pathogens and lay the foundation for developing new antifungal agents.

## 2. Results and Discussion

### 2.1. Chemical Composition of Apolar Secondary Metabolites of C. morifolium Leaves

A total of 13 cultivars of *C. morifolium* (Table 1) commonly used for making chrysanthemum tea were selected for this study. To simplify cultivar references between the main text and figures/tables, these 13 cultivars were coded as Cm1 to Cm13 (Table 1). To avoid variations due to different developmental stages, the third leaf from the top of individual mature plants for each of the cultivars was collected and subjected to organic extraction. All extracts were then analyzed using gas chromatography–mass spectrometry (GC-MS). The number of apolar secondary metabolites detected from the 13 cultivars ranged from 15 to 26 (Appendix A). In total, 37 apolar secondary metabolites were detected in three chemical classes: terpenoids, fatty acid derivatives, and benzenoids. Leaves of the cultivar ‘Xiao xiang ju’ (Cm 10) exhibited the highest concentration of total apolar secondary metabolites (982.35 ± 89.08 μg g^−1^ fresh weight), followed by the cultivars ‘Wan gong ju’ (Cm 9) (947.08 ± 90.28 μg g^−1^ fresh weight) and ‘Xiao huang ju’ (Cm 10) (770.31 ± 211.56 μg g^−1^ fresh weight) (Figure 1).

A total of 26 terpenoid components, accounting for 60.57–98.42% of the total apolar secondary metabolites among different cultivars, were identified in the 13 cultivars (Figure 2). The highest concentration of terpenoids was observed in ‘Wan gong ju’ (Cm 9) (829.01 ± 91.30 μg g^−1^ fresh weight) and the lowest concentration in ‘Huang xiang li’ (Cm 6) (167.16 ± 18.03 μg g^−1^ fresh weight). It has been documented that terpenoids (e.g., β-pinene, eucalyptol, camphor, borneol, and bornyl acetate) are usually the major constituents of essential oils in a wide range of *C. morifolium* cultivars [15,16,17,18,19], which is consistent with the results in our study. For 26 individual terpenoid compounds in 13 cultivars, eucalyptol, β-ylangene, *epi*-bicyclosesquiphellandrene, (*E*)-β-farnesene, and α-curcumene were identified from all the 13 cultivars, while longifolenaldehyde and β-selinene were identified only in ‘Chuju’ (Cm 1) Appendix A).

Fatty acid derivatives detected in the leaf extracts accounted for 1.58% to 39.43% of the total apolar secondary metabolites. Interestingly, the total fatty acid derivatives concentration of ‘Xiao xiang ju’ (Cm 4) (387.34 ± 35.13 μg g^−1^ fresh weight) was significantly higher than those in other cultivars (9.65–187.44 μg g^−1^ fresh weight). Benzenoids were detected only in ‘Jin si huang ju’ (Cm 2) (53.35 ± 41.05 μg g^−1^ fresh weight) and ‘Xiao xiang ju’ (Cm 4) (25.48 ± 3.86 μg g^−1^ fresh weight).

### 2.2. PCA and HCA Analysis Based on the Terpenoid Contents

Since terpenoids represent the dominant class of apolar secondary metabolites in most cultivars examined, principal component analysis (PCA) and hierarchical clusters analysis (HCA) based on the terpenoid contents in the leaf extracts were conducted to define the relationship of these 13 cultivars. The chemical PCA horizontal axis explained 26.10% of the total variance, while the vertical axis explained a further 18.14% (Figure 3A). The HCA based on the Euclidean distance between groups placed the 13 cultivars into 3 groups (I, II, and III) (Figure 3B).

Group I contained three cultivars (‘Xiao xiang ju’ (Cm 4), ‘Xiao huang ju’ (Cm 10), and ‘Wan gong ju’ (Cm 9)) and was isolated as a separate group in PCA (Figure 4A) and as a deep dichotomy in HCA analysis (Figure 3B). In this group, the leaf extracts were characterized by high levels of zingiberene (184.02 ± 35.2 μg g^−1^ fresh weight, 164.07 ± 92.97 μg g^−1^ fresh weight, and 204.07 ± 96.78 μg g^−1^ fresh weight, respectively) and β-sesquiphellandrene (119.43 ± 13.50 μg g^−1^ fresh weight, 18.97 ± 4.90 μg g^−1^ fresh weight, and 129.61 ± 5.68 μg g^−1^ fresh weight, respectively). The PCA results also indicated that the three cultivars in this group could be differentiated by the presence of some other compounds such as β-ylangene (62.83 ± 6.36 μg g^−1^ fresh weight, 47.85 ± 14.53 μg g^−1^ fresh weight, and 95.04 ± 5.99 μg g^−1^ fresh weight, respectively) (Figure 3A).

Group II also contained three cultivars (‘Da yang ju’ (Cm 5), ‘She yang hong xin ju’ (Cm 12), and ‘Su ju 9’ (Cm 11)), whose extracts were specialized in high concentrations of α-curcumene (148.68 ± 8.55 μg g^−1^ fresh weight, 156.06 ± 23.80 μg g^−1^ fresh weight, 126.57 ± 24.09 μg g^−1^ fresh weight, respectively) and β-ylangene (141.42 ± 5.38 μg g^−1^ fresh weight, 131.12 ± 7.22 μg g^−1^ fresh weight, 95.72 ± 27.17 μg g^−1^ fresh weight, respectively).

Group III, which was the largest group, contained seven cultivars (‘Huang xiang li’ (Cm 6), ‘Qi yue bai’ (Cm 7), ‘Huang ju’ (Cm 8), ‘Jin si huang ju’ (Cm 2), ‘Chu ju’ (Cm 1), ‘Bai xiang li’ (Cm 13), and ‘Hang bai ju’ (Cm 3)), whose extracts were characterized by high concentrations of the diterpenoid phytol (23.55 ± 13.78 μg g^−1^ fresh weight, 27.08 ± 10.27 μg g^−1^ fresh weight, 17.65 ± 2.85 μg g^−1^ fresh weight, 14.69 ± 14.34 μg g^−1^ fresh weight, 23.85 ± 2.81 μg g^−1^ fresh weight, 45.81 ± 27.39 μg g^−1^ fresh weight, and 18.16 ± 8.70 μg g^−1^ fresh weight, respectively). The PCA results also suggested the cultivars in this group could be differentiated by the presence of some sesquiterpenoids, such as (*E*)-β-farnesene (6.06–10.49 μg g^−1^ fresh weight), *epi*-bicyclosesquiphellandrene (7.71–14.91 μg g^−1^ fresh weight), and (*S*,1Z,6Z)-8-isopropyl-1-methyl-5-methylenecyclodeca-1,6-diene (5.47 ± 0.91 μg g^−1^ fresh weight). Moreover, the cultivar ‘Hang bai ju’ was characterized by relatively high concentrations of zingiberene (111.26 ± 7.00 μg g^−1^ fresh weight), (*E*)-β-caryophyllene (41.12 ± 5.13 μg g^−1^ fresh weight), β-elemene (18.12 ± 4.74 μg g^−1^ fresh weight), and α-farnesene (22.72 ± 5.45 μg g^−1^ fresh weight). As an exception, ‘Bai xiang li’ (Cm 13) was separated from the other cultivars by relatively high concentrations of oxygenated terpenes including filifolone (16.53 ± 1.95 μg g^−1^ fresh weight), (*E*)-ocimenone (10.34 ± 1.06 μg g^−1^ fresh weight), and isopiperitenone (19.02 ± 1.95 μg g^−1^ fresh weight). In this group, 8,9-epoxyacorenon-B was absent in all cultivars. In contrast, longifolenaldehyde was detected only in ‘Chu ju’ (Cm 1) and limonene was detected only in ‘Huang ju’ (Cm 8) and ‘Qi yue bai’ (Cm 7). Collectively, important qualitative and quantitative variations in the contents of terpenoids were observed among different cultivars of *C. morifolium*. It will be an interesting future research direction to uncover the molecular and biochemical mechanisms underlying such chemical variations. Terpene synthases are pivotal enzymes catalyzing terpenoid biosynthesis [20]. Divergence in catalytic functions and expression regulation of individual terpene synthase genes may play a critical part in terpenoid variations among the cultivars of *C. morifolium*.

### 2.3. Antifungal Activity of Apolar Secondary Metabolites from C. morifolium Leaves

Many apolar secondary metabolites detected from *C. morifolium*, especially terpenoids, are known to have antimicrobial activities [21,22,23,24,25]. To evaluate the antimicrobial properties of the apolar secondary metabolites from *C. morifolium* leaves, we tested three species of pathogenic fungi, *Fusarium oxysporum*, *Magnaporthe oryzae*, and *Verticillium dahliae*. *F. oxysporum* exhibits variations in infecting *C. morifolium* and other species of *Chrysanthemum* [26,27]. Identifying metabolites with antifungal activities may provide a chemical interpretation for the varied resistance or tolerance. *M. oryzae* and *V. dahliae* are both important fungal pathogens causing enormous economic losses and model fungal pathogens [28,29,30,31]. *M. oryzae* is best known as the causal agent of the rice blast disease. *V. dahliae*, a soil-borne pathogen found in temperate and subtropical zones, infects both herbaceous and woody host plants, especially the *Solanaceae*: tobacco, potatoes, peppers, and eggplant. Screening effective antagonistic chemicals would be of great economic significance for managing diseases caused by these and other fungal pathogens.

Apolar secondary metabolites extracted from the leaves of the 13 cultivars of *C. morifolium* exhibited differences in their antifungal properties against the three species of fungi (Figure 4). Among the tested cultivars, the extracts of four cultivars (‘Jin si huang ju’ (Cm 2), ‘Huang xiang li’ (Cm 6), ‘Huang ju’ (Cm 8), and ‘Bai xiang li’ (Cm 13)) showed significant inhibitory activity mainly on the growth of *F. oxysporum* (Figure 4A), whereas the extracts of nine cultivars (‘Chu ju’ (Cm 1), ‘Jin si huang ju’ (Cm 2), ‘Xiao xiang ju’ (Cm 10), ‘Huang xiang li’ (Cm 6), ‘Huang ju’ (Cm 8), ‘Wan gong ju’ (Cm 9), ‘Xiao huang ju’ (Cm 4), ‘Su ju 9’ (Cm 11), and ‘Bai xang li’ (Cm 13)) showed significant inhibitory activity mainly on the growth of *M. oryzae* (Figure 4B). The extracts of nine cultivars (‘Chu ju’ (Cm 1), ‘Jin si huang ju’ (Cm 2), ‘Hang bai ju’ (Cm 3), ‘Xiao xiang ju’ (Cm 4), ‘Huang xiang li’ (Cm 6), ‘Qi yue bai’ (Cm 7), ‘Huang ju’ (Cm 8), ‘She yang hong xin ju’ (Cm 12), and ‘Bai xiang li’ (Cm 13)) showed significant inhibitory activity mainly on the growth of *V. dahliae* (Figure 4C). Among these cultivars, ‘Huangju’ (Cm 8) showed the strongest inhibitory activity against all three species of fungi (Figure 4D). Our results point to a possible application of *Chrysanthemum* leaves as a source of fungicide for the control of certain fungal pathogens such as *F. oxysporum*, *M. oryzae*, and *V. dahliae*. To this end, it will be important to compare the antifungal activities of *Chrysanthemum* leaf extracts with those of comparable commercial products.

### 2.4. PCA and HCA Analysis Based on the Antifungal Effect

PCA coupled with HCA based on the values of fungal colony diameters was conducted to evaluate the correlation between the apolar secondary metabolites and the antifungal activities of the extracts made from the 13 cultivars. The PCA horizontal axis explained 70.16% of the total variance, while the vertical axis explained a further 19.03% (Figure 5A). The HCA based on the Euclidean distances between groups indicated that all the cultivars can be classified into four groups (“a”, “b”, “c”, and “d”) according to their colony diameter. Both the PCA (Figure 5A) and the HCA (Figure 5B) analysis showed that Group a was composed of ‘Huang ju’ (Cm 8) and ‘Huang xiang li’ (Cm 6), with smallest colony growth diameter, indicating the strongest antifungal activity of these two cultivars. Group b was represented by ‘Hang bai ju’ (Cm 3), ‘Qi yue bai’ (Cm 7), ‘Da yang ju’ (Cm 5), ‘She yang hong xin ju’ (Cm 12), and the control. Being opposite to Group a, Group b was considered as the most inactive against the studied fungus species. None of the cultivars showed significant inhibitory activity on *F. oxysporum* and *M. oryzae*. However, among these cultivars, ‘Hang bai ju’ (Cm 3), ‘Qi yue bai’ (Cm 7), and ‘She yang hong xin ju’ (Cm 12) had significant inhibition activity on *V. dahliae*. Group c was composed of three cultivars, ‘Chu ju’ (Cm 1), ‘Xiao xiang ju’ (Cm 4), and ‘Jin si huang ju’ (Cm 2), whose extracts displayed significant antifungal activities against *M. oryzae* and *V. dahliae*. Group d was represented by the remaining four cultivars (‘Bai xiang li’ (Cm 13), ‘Wan gong ju’ (Cm 9), ‘Su ju 9’ Cm 11), and ‘Xiao huang ju’ (Cm 4)), whose extracts showed strong inhibitory activities against *M. oryzae*. Among these four cultivars, the extract of ‘Bai xiang li’ (Cm 13) exhibited an inhibitive effect on the growth of all three species of fungi, while ‘Wan gong ju’ (Cm 9), ‘Su ju 9’ (Cm 11), and ‘Xiao huang ju’ (Cm 10) showed the inhibitive effect only to *M. oryzae.* Many terpenoid compounds identified in *C. morifolium* leaves have been reported to have an antifungal effect on various species of fungi [21,22,23,24,25]. By comparing the PCA/HCA results based on terpenoid contents and antifungal activities, the antifungal effects of several cultivars (including ‘Da yang ju’ (Cm 5), ‘Huang xiang li’ (Cm 6), ‘Huang ju’ (Cm 8), ‘She yang hong xin ju’ (Cm 12)) could be associated with several terpenoids, including zingiberene, β-sesquiphellandrene, α-curcumene, (*E*)-β-farnesene, and α-farnesene. To establish causal correlation between these individual terpenes and antifungal activities will be interesting future research.

### 2.5. Conclusions

In this study, we showed both similarities and differences in chemical composition of apolar secondary metabolites in leaves among 13 cultivars of *C. morifolium*. With terpenoids being the dominant class of secondary metabolites, some terpenoids occur in all these cultivars (Appendix A), suggesting conserved biosynthesis and biological functions. Nonetheless, there are also large variations in the qualities and quantities of terpenoids among these cultivars (Appendix A). Because these plants were cultivated under the same optimal conditions, the variations are most likely genetically determined, probably reflecting both the rapid evolution of the terpene biosynthetic pathway and the complex domestication/breeding history of *C. morifolium* [20,32]. Through the glimpse of this study, enormous variations in terpene chemistry can be expected to be discovered among the thousands of cultivars of *C. morifolium*. Such vast diversity of terpenoid chemistry could be explored for various applications. As revealed in this study, terpenoids of *C. morifolium* may be used as antifungal agents (Figure 4 and Figure 5). PCA and HCA analyses collectively showed that antifungal properties of apolar secondary metabolites of *C. morifolium* leaves could be partly attributed to the contents of some terpenoids, including zingiberene, β-sesquiphellandrene, α-curcumene, (*E*)-β-farnesene, α-farnesene, and 1,8-cineole. This information could be used to facilitate the breeding of new cultivars of *C. morifolium* with enhanced resistance to fungal pathogens and lay a foundation for developing new terpenoid-based antifungal agents.

## 3. Materials and Methods

### 3.1. Plant Material

All 13 cultivars of *C. morifolium* were cultivated in the *Chrysanthemum* Germplasm Resource and Preservation Center, Nanjing Agricultural University, China (118°98′ N, 32°07′ E). All plants were grown in the greenhouse under the same conditions (at 27–32 °C; with light intensity at 700–1100 lux; humidity at 75%–87%). The third leaf from the top for individual plants at the reproductive stage (full blooming) for each cultivar was collected and subjected to organic extraction.

### 3.2. Organic Extraction of Leaves of C. morifolium

For organic extraction, leaf tissue of each cultivar was ground into powder and ethyl acetate of HPLC grade (Macklin Technology, Shanghai, China) was added as solvent in 5:1 (volume to weight) ratio. After shaking at room temperature for 2 h and a subsequent centrifugation (5000 r/min, 5 min), the organic phase was collected for GC-MS analysis and antifungal bioassays. For preparation of samples for GC-MS analysis, nonyl acetate (CAS:143-13-5, ≥98%, Sigma Aldrich, St Louis, MO, USA) was added to ethyl acetate (0.002%) as an internal standard. For each cultivar, three biological replicates were analyzed.

### 3.3. GC-MS Analysis and Identification of Extract Constituents

The analyses were performed using a GC-MS system (Agilent Intuvo 9000 GC system coupled with an Agilent 7000D Triple Quadrupole mass detector). Separation was performed on an Agilent HP 5 MS capillary column (30 m × 0.25 mm) with helium as carrier gas (1 mL·min^−1^ of flow rate). The injection volume of each sample was 1 µL. The temperature of the injection port was 260 °C, with a split mode (split ratio = 5:1). The column temperature program of gradient heating was adopted as follows: The temperature was initiated at 40 °C, followed by an increase to 250 °C at a rate of 5 °C/min. The MS conditions included an EI ion source temperature of 230 °C, an ionization energy of 70 eV, and a mass scan range of 40–500 amu. The separated constituents were identified by comparing their mass spectra with the authentic standards or those in the NIST17 MS library (National Institute of Standards and Technology). Retention indices were calculated using a series of C7 to C40 hydrocarbon standard (Sigma-Aldrich, St. Louis, MO, USA). Each constituent was quantified based on the comparison of its peak area with that of the internal standard, and the contents were expressed as μg g^−1^ fresh weight. Minor peaks, which were defined as those having a peak area less than 1% of total peak area, were excluded from analysis.

### 3.4. Assessment of Antifungal Activity

Three pathogenic fungi, *Magnaporthe oryzae*, *Verticillium dahliae*, and *Fusarium oxysporum*, were tested in this study. *M. oryzae* strain Guy11, originally isolated from a rice field, is a model strain for studying rice blast disease [33]. *V. dahliae* strain V991 was originally isolated from a cotton field [34]. *F. xysporum* strain CFD-B2 was isolated from the cut flower chrysanthemum ‘Shenma’ plant in the chrysanthemum experimental station of Nanjing Agricultural University in the summer of 2016. These three species of pathogenic fungi were activated and the agarose plug with mycelium was cut from the edge of the original fungus culture under sterile conditions. Each plug was inverted and the mycelium side placed in contact with the medium in the center of dishes containing PDA, and cultured at 28 °C for five days in an incubator to carry out experiments. Ethyl acetate extracts were screened for antifungal activity in vitro by measurement of inhibitory zone diameter as previously described [21]. In order to investigate the antifungal activity of the extracts, the modified mycelial growth test with malt agar was used [35]. Petri dishes (90 mm in diameter) containing sterilized PDA media were used for the test and a 6 mm plug of mycelial agar, obtained from the edge of three-day-old cultures of fungi, was transferred to the center of each Petri dish. For each Petri dish, based on our preliminary experiment, 200 μL of the extract could cover the entire agar plate and was chosen as the volume of extract in each assay. Also performed were assays with 200 μL of ethyl acetate, the organic solvent used for leaf extraction, which served as a negative control. After the plates were cultured for 5 d at 28 °C, the diameter (mm) of the colony zone was determined with a caliper. All of the experiments were performed in triplicates with each replicate containing 10 plates.

### 3.5. Statistical Analysis

The differences in antifungal activity between extracts from the 13 cultivars and control were analyzed using Student’s *t*-test, and the significance of the differences was determined at *p* < 0.05. Principal component analysis (PCA) and hierarchical cluster analysis (HCA) using SPSS 22.0 software were conducted to determine relationships among the 13 cultivars in their chemical composition and antifungal activities based on the concentrations of individual terpenoids and the values of fungi growth diameters, respectively. The top two highest principal components (PC1 and PC2) were chosen to construct a loading diagram, according to the factor scores. The tested samples were positioned in the two-dimensional space, with some obvious groupings. HCA based on the squared Euclidean distance and the method of between-group linkages was used to cluster the samples with different relative terpenoid contents and antifungal effect.

## Figures and Tables

**Figure 1 molecules-24-04202-f001:**
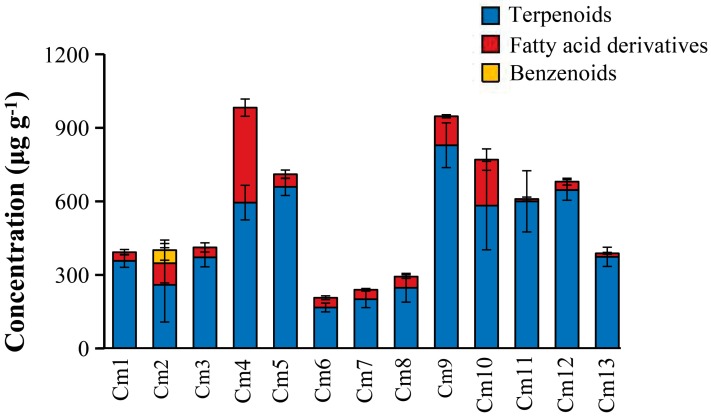
Concentrations of three classes (terpenoids, fatty acid derivatives, and benzenoids) of apolar secondary metabolites in leaf extracts of 13 cultivars of *C. morifolium*. Cm 1–13 refer to the cultivar codes in Table 1.

**Figure 2 molecules-24-04202-f002:**
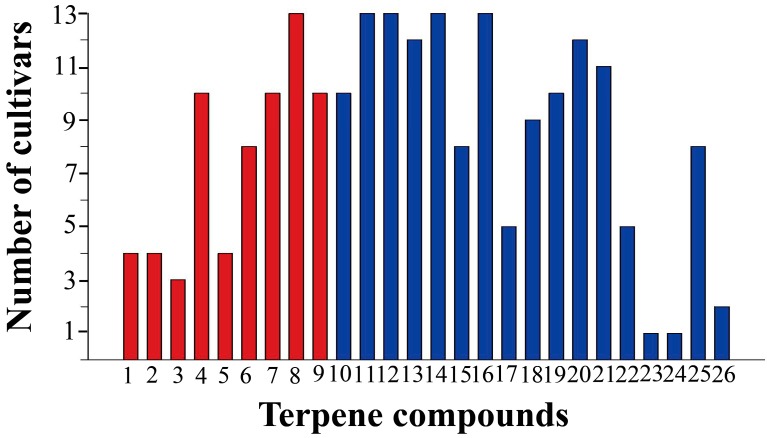
Occurrence of 26 individual terpenoid compounds in 13 cultivars of *C. morifolium*. Red bars represent monoterpenoids, while blue bars represent sesquiterpenoids. Numbers 1–26 correspond to the compound numbers listed in Appendix A.

**Figure 3 molecules-24-04202-f003:**
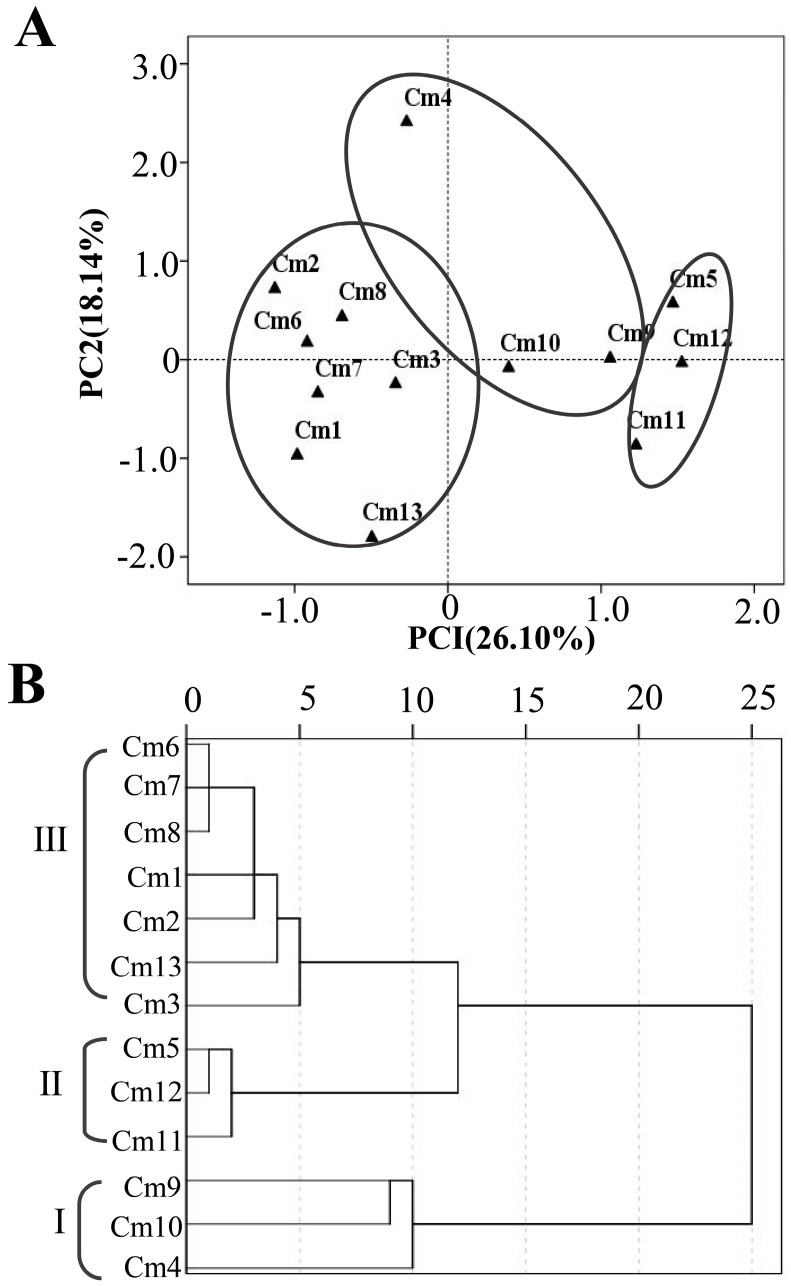
(**A**) PCA analysis of the detected apolar secondary metabolites composition of leaf extract from 13 cultivars of *C. morifolium*. (**B**) Dendrogram obtained by hierarchical cluster analysis based on the Euclidean distance between groups of the main chemical composition of leaf extracts of 13 tea *Chrysanthemum* cultivars. Cm 1–13 refer to the cultivar codes in Table 1.

**Figure 4 molecules-24-04202-f004:**
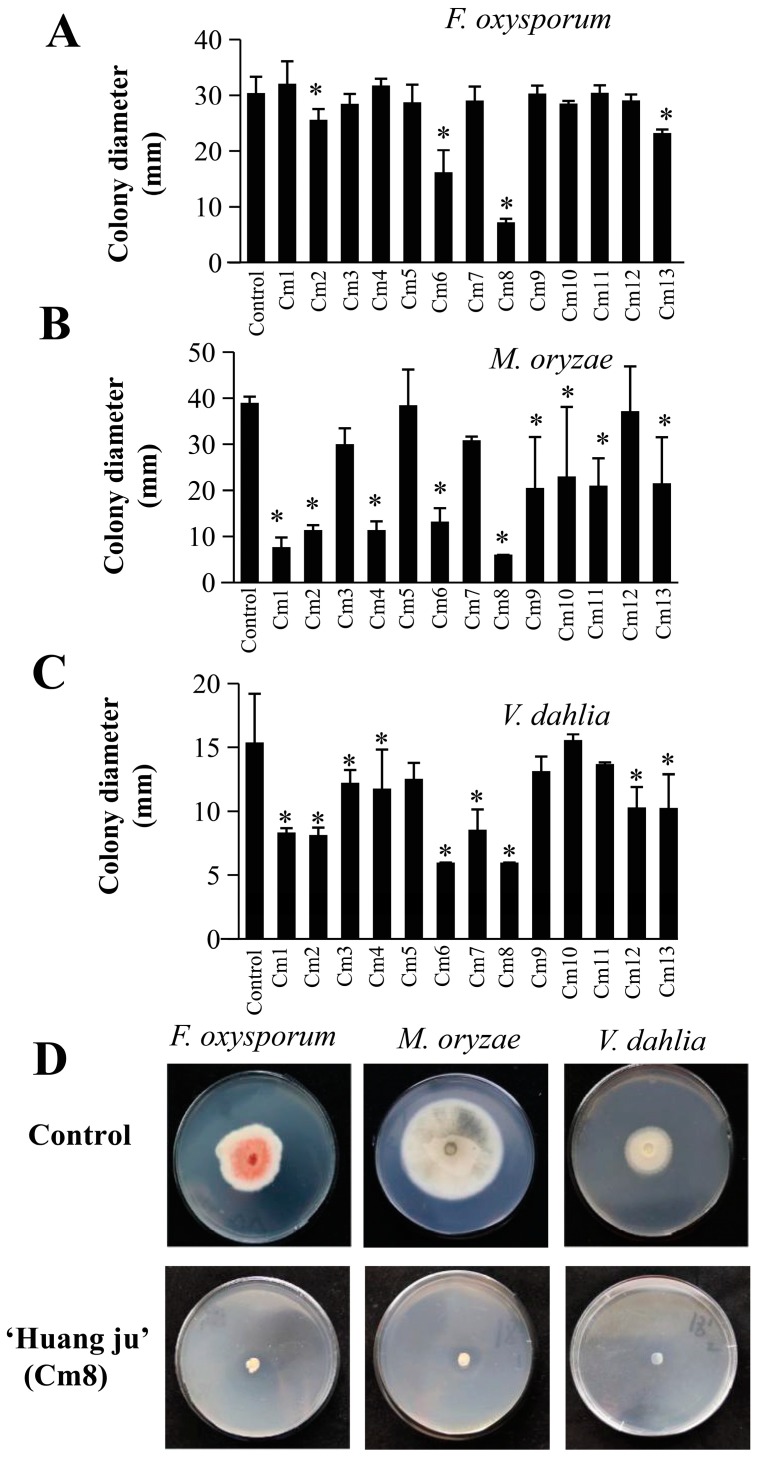
Effect of apolar secondary metabolites from 13 cultivars of *C. morifolium* on the growth of three pathogenic fungi: *Fusarium oxysporum* (**A**), *Magnaporthe oryzae* (**B**), and *Verticillium dahliae* (**C**). Cm 1–13 refer to the source of apolar secondary metabolites as listed in Table 1. Control refers to the treatment with organic solvent ethyl acetate, which served as a negative control. Data were presented as means ± standard deviations based on three replicates. * indicates significance at the statistical level (*p* < 0.05) compared to the control. (**D**) Representative growth of *Fusarium oxysporum*, *Magnaporthe oryzae*, and *Verticillium dahlia* treated with a control (ethyl acetate) or with an extract made from ‘Huang ju’ (Cm 8) leaves.

**Figure 5 molecules-24-04202-f005:**
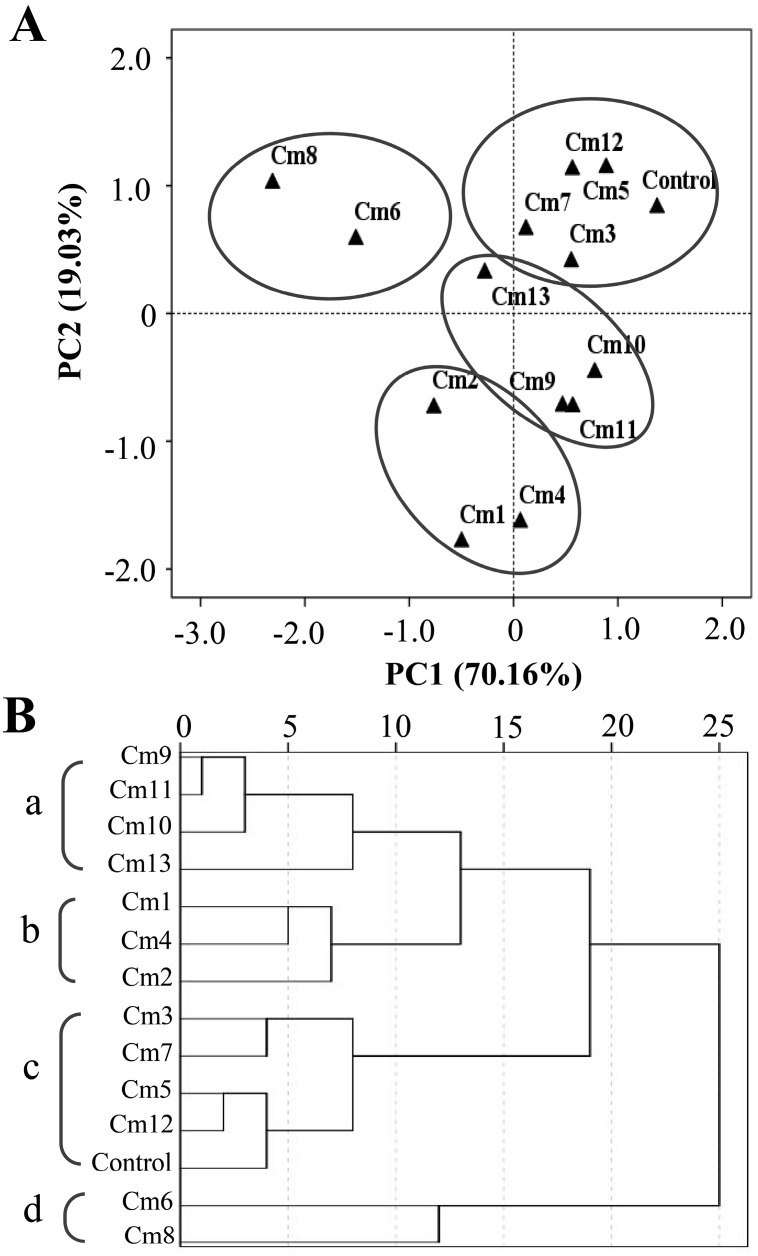
(**A**). PCA analysis of the antifungal activity of leaf extracts made from 13 cultivars of *C. morifolium* against the three fungal species *Fusarium oxysporum*, *Magnaporthe oryzae,* and *Verticillium dahliae*. (**B**). Dendrogram obtained by hierarchical cluster analysis based on the Euclidean distance between groups of the antifungal activities of leaf extracts of 13 cultivars of *C. morifolium*. Cm 1–13 refer to the cultivar codes in Table 1.

**Table 1 molecules-24-04202-t001:** Cultivars of *C. morifolium* used in this study.

Code	Cultivar	Collection Locality
Cm1	Chuju	Nanjing, Jiangsu province, China
Cm2	Jin si huang ju	Nanjing, Jiangsu province, China
Cm3	Hang bai ju	Nanjing, Jiangsu province, China
Cm4	Xiao xiang ju	Nanjing, Jiangsu province, China
Cm5	Da yang ju	Nanjing, Jiangsu province, China
Cm6	Huang xiang li	Nanjing, Jiangsu province, China
Cm7	Qi yue bai	Nanjing, Jiangsu province, China
Cm8	Huang ju	Nanjing, Jiangsu province, China
Cm9	Wan gong ju	Nanjing, Jiangsu province, China
Cm10	Xiao huang ju	Nanjing, Jiangsu province, China
Cm11	Su ju 9	Nanjing, Jiangsu province, China
Cm12	She yang hong xin ju	Nanjing, Jiangsu province, China
Cm13	Bai xiang li	Nanjing, Jiangsu province, China

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
