# Peer review of "Characterization of Composition and Antifungal Properties of Leaf Secondary Metabolites from Thirteen Cultivars of Chrysanthemum morifolium Ramat"

_molecules, 2019, doi:10.3390/molecules24234202_

Round 1
Reviewer 1 Report
In the manuscript entitled “Characterization of Composition and Antifungal Properties of Leaf Secondary Metabolites from Thirteen Cultivars of Chrysanthemum morifolium ramat.”, the authors evaluated of the chemical composition profile of apolar secondary metabolites from leaves of 13 different cultivars of C. morifolium and to determine whether these compounds have any antifungal effect. The work is overall well done, carefully thought and performed and the manuscript is well written and easy to read, follow and the data presented support the author's conclusions. All experimental methods are well explained. Other Specific comments:
Please insert a new paragraph showing biotic and abiotic to explain the chemical profile variations of these cultivars and your importance as a potential antifungal. I suggest cite the paper:
de Macêdo, Delmacia G., et al. "Effect of seasonality on chemical profile and antifungal activity of essential oil isolated from leaves Psidium salutare (Kunth) O. Berg." PeerJ 6 (2018): e5476.
An important result was the comparison of the effects of essential oil of all cultivar. Exist significant difference in antifugal activity for same fungal and same concentrations?
No statistical analysis is present in this work. This analysis is required to show a significant difference in the activity of essential oil of the cultivars compared to control. No antifungal (positive control) was used in antifungal tests. This use is necessary to determine the effectiveness
The chemical profile quantifications are determinate by the external standard method in the Mass spectrometry method? The validation of the method and the quantification must be performed following ICH procedures (ICH, 2012). I would like underline that this aspect is very important to answer to the requirements of the journal. The authors need to determine the limit of detection (LOD) and limit of quantification (LOQ) used. The chemical analysis no mention, those retention indices (RI) were calculated using a series of n-alkanes (C7-C33). The analytical part of your work does not comply with major requirements, however, identification of GC-eluted constituents of the essential oil must involve a strict comparison of BOTH MS data and Linear Retention Indices (LRIs), not times, with data recorded for authentic compounds under similar conditions. Then, the GC-FID is a technical more indicate, please show information’s that the identifications of compounds no show similar conditions of analysis
The experimental methods fail to demonstrate important information as concentrations used, dissolutions and evaluation of inhibition.
What is the criterion used to selections of the doses used in the antifungal assay? Explain. Utilized reference for selections of dose.
No statistical analysis was observed in figures 2 and 3. This point is important for the analysis of the results.
The quality of English writing throughout the manuscript is substandard, needless to list all the errors and unprofessional expressions. Assistance from native or professional English writer may be needed
Author Response
Review 1
In the manuscript entitled “Characterization of Composition and Antifungal Properties of Leaf Secondary Metabolites from Thirteen Cultivars of Chrysanthemum morifolium ramat.”, the authors evaluated of the chemical composition profile of apolar secondary metabolites from leaves of 13 different cultivars of C. morifolium and to determine whether these compounds have any antifungal effect. The work is overall well done, carefully thought and performed and the manuscript is well written and easy to read, follow and the data presented support the author's conclusions. All experimental methods are well explained. Other Specific comments:
We thanks the positive evaluations!
Please insert a new paragraph showing biotic and abiotic to explain the chemical profile variations of these cultivars and your importance as a potential antifungal. I suggest cite the paper: de Macêdo, Delmacia G., et al. "Effect of seasonality on chemical profile and antifungal activity of essential oil isolated from leaves Psidium salutare (Kunth) O. Berg." PeerJ 6 (2018): e5476.
Response: Because the plants of 13 cultivars in this study were grown under the same optimal conditions, the variations in chemical profiles are not due to biotic or abiotic stresses. Instead, this is due to the difference in genetics. This point is emphasize in Conclusions. The suggested reference is cited.
An important result was the comparison of the effects of essential oil of all cultivar. Exist significant difference in antifugal activity for same fungal and same concentrations?
Response: Because of the differences in the number of apolar secondary metabolites and their respective concentrations among cultivars, it is impossible to define the “same concentrations”. Instead, we compared the different cultivars and statistical analysis was presented in Figure 4.
No statistical analysis is present in this work. This analysis is required to show a significant difference in the activity of essential oil of the cultivars compared to control. No antifungal (positive control) was used in antifungal tests. This use is necessary to determine the effectiveness
Response: Statistical analysis was conducted to analyze the difference in antifungal activity of different cultivars in comparison to that of mock (negative) control. We were comparing different cultivars. It was not objective to comparing chrysanthemum extracts to a known commercial antifungal product. Certainly, there must be many antifungal products on the market. It is hard to justify which one to choose and at what concentrations, etc. for comparison.
The chemical profile quantifications are determinate by the external standard method in the Mass spectrometry method? The validation of the method and the quantification must be performed following ICH procedures (ICH, 2012). I would like underline that this aspect is very important to answer to the requirements of the journal. The authors need to determine the limit of detection (LOD) and limit of quantification (LOQ) used. The chemical analysis no mention, those retention indices (RI) were calculated using a series of n-alkanes (C7-C33). The analytical part of your work does not comply with major requirements, however, identification of GC-eluted constituents of the essential oil must involve a strict comparison of BOTH MS data and Linear Retention Indices (LRIs), not times, with data recorded for authentic compounds under similar conditions. Then, the GC-FID is a technical more indicate, please show information’s that the identifications of compounds no show similar conditions of analysis
Response: It is certainly true that more rigorous approaches could help with the chemical aspects of this study. However, for this biology-orientated study, as many other similar works, quantifications using external standard method is generally accepted. It is not the objective of this study to determine absolute concentrations of individual compounds, which is almost impossible, but our objective to compare the questions under same conditions. Retention indices (RI) were calculated using a series of n-alkanes (C7-C40). This information was provided in the note to Table 2 (Now as Table S1 in the revised version). We moved this information to Methods for better visibility.
The experimental methods fail to demonstrate important information as concentrations used, dissolutions and evaluation of inhibition.
Response: Our purpose is to compare a relatively large number of cultivars, not focusing on a single cultivar. Evaluations of the different concentrations, dissolutions and inhibition for single cultivars will be important future research.
What is the criterion used to selections of the doses used in the antifungal assay? Explain. Utilized reference for selections of dose.
Response: Justification is now provided in Method.
No statistical analysis was observed in figures 2 and 3. This point is important for the analysis of the results.
Response: In Figure 3, the PCA was conducted based on the average value of the colony diameter, which has been gone through statistical treatment. Figure 2 is not quantification. It is a summary on the presence/absence of each terpene in each cultivar. Statistical analysis is unnecessary.
The quality of English writing throughout the manuscript is substandard, needless to list all the errors and unprofessional expressions. Assistance from native or professional English writer may be needed
Response: The entire manuscript has been carefully checked for English quality.
Reviewer 2 Report
As in the attached file.

Author Response
Review 2
INTRODUCTION chapter lacks information on chemical composition of C. morifolium based on current literature.
Responses: Some additional references on chemical composition of C. morifolium was added.
The authors should also add information on medicinally used raw materials (it is mentioned in the manuscript that it is drunk as a tea) originated from this plant, and the direction of its application. It should be stressed that it is one of the oldest plants described in Chinese Materia Medica.
Responses: The medicinal use of C. morifolium is added, including its description in the first major work dedicated to Chinese Materia Medica known as Shennong Bencao Jing.
morifolium ramat. – should be Ramat.
Responses: Thanks, the name has been corrected.
MATERIALS AND METHODS chapter 3.1 – In the abstract section there is an information that for the experiment 13 cultivars were selected - on the basis of which data the selection was made. This should be mentioned in this chapter.
Responses: The basis of cultivar selection has been elaborated. These cultivars have been traditionally cultivated to be used as teas, making them comparable.
The authors write that “To avoid variations due to different developmental stages, the third leaf from the top of an individual mature plants for each of the cultivars was collected”, however, from this information, and from Materials and Methods chapter the reader does not know what was the developmental phase of the plants. Please introduce proper data, i.e.: What was the age of plants, and what was their developmental stage (vegetative or generative stage?). How many plants were grown per cultivar. How many leaves were collected for the experiment (it is not clear, what are biological replicates?). Were the leaves dried? Is so, how was the drying process carried out.
Responses: The detailed information about the age and developmental stage were added in the ‘material’ part.
Chapter 3.4 Where from the pathogenic fungi originated (were purchased?). How their identity was confirmed. How they were grown for the experiment.
Responses: The information about the origination and identification of the fungi was added.
In the ABSTRACT there should be an information that antifungal activity was investigated in in vitro conditions.
Responses: It was emphasized in Abstract that antifungal activity was investigated in in vitro conditions as suggested.
In this chapter (Materials and Methods) a table should be created to name all the synonyms for cultivars (instead of giving their names at the end of Tables or Figures in the Results chapter). Some additional information concerning the cultivars should be introduced in such table, i.e. their accession numbers and origin.
Responses: A new Table 1 was added to streamline the description of cultivars particular in all figures.
RESULTS AND DISCUSSION chapter: Table 1 is not needed, this data can be read from Table 2.
Responses: The previous Table 1 was replaced by new one as above mentioned.
In Tab. 2 – why some names are written with capital letters and other with small letters. This should be harmonized.
Responses: The compounds name have been harmonized as the small letters uniformly.
In the first line instead of “Concentrations” should be “cultivars”.
Responses: “Concentrations” was replaced by “cultivars”.
Units should be given at the end of Tab 2 title, i.e. Apolar secondary metabolites identified in the leaf extracts of 13 cultivars of C. morifolium (μg g-1 ).
Responses: Units was added as suggested.
Figure 1 and Figure 2 – these are also repeated information from table 2. It should be removed from the paper.
Responses: According to the suggestions of all reviewers, we elected to change Table 2 to a supplemental document. As such, Figure 1 and Figure 2 are important in the main text to introduce our results.
REFERENCE chapter should be checked for the style. The article requires correction of a native speaker who has expertise in the thematic scope of the article.
Responses: References style have been checked. The entire manuscript has been carefully checked by English quality.
Reviewer 3 Report
The authors present an interesting study that looks at anti-fungal activity of apolar compounds extracted from Chrysanthemum leaves. I have a few concerns that I have listed below.
Major concerns:
----- ---------
Tables 1 and 2: the numbers do not match, e.g. In the Benzenoids category, C4 has 0 listed in Table 1, whereas, C4 has 1 value in Table 2.
In the text, cultivar names are being used, whereas, Table 1 explicitly sets up code names for all the cultivars. Please stick to one. Once the code name has been finalized, use that throughout the text.
Change the code names for the cultivars as C1 to C13 might be mistaken for the number of carbon atoms in complex carbon-based molecules. The authors should use different initials to represent the genera and species, like 'Chmor' or 'Cmo'.
Figure legends: Please do not list all the names for C1-C13. The authors could direct the reader to refer to Table 1, as was done for the compounds in Fig. 2 legend.
Figure 2: The frequency ranges from 1 - 13, which should be easy to show instead of using percentage on Y-axis.
Figure 3B: The use of bracket lines to show groupings in the HCA does not match with the PCA. Please fix the figure.
Lines 144-185: It is very descriptive. It would be really interesting if the authors could show that the amounts of some of these apolar compounds are statistically/significantly different between the three groups based on PCA/HCA.
What is the basis of these differences? Is it genetic, population structure? The authors mention this in one line in the conclusions. Please expand on it a bit more.
The fungal isolates tested in this study, where were those obtained from? Were those isolated from Chrysanthemum?
Figure 4D: What is CK? Please add to foot-note. In methods, only two artificial media, PDA and Malt extract, are mentioned.
Lines 213-215: The name of the cultivar mentioned in the text and that shown in Fig. 4D does not match.
Lines 293-305: How many plates were used per replicate?
In the HCA figures, Fig. 3B, Fig. 5B: What is the meaning of the dotted line?
Minor comments:
----- ---------
Line 102: typo 'apola'
Make Table 2 supplementary
Author Response
Review 3
Major concerns:
----- ---------
Tables 1 and 2: the numbers do not match, e.g. In the Benzenoids category, C4 has 0 listed in Table 1, whereas, C4 has 1 value in Table 2.
Response: Table 1 was removed and Table 2 changed to a supplementary table.
In the text, cultivar names are being used, whereas, Table 1 explicitly sets up code names for all the cultivars. Please stick to one. Once the code name has been finalized, use that throughout the text.
Response: The code names have been used throughout the text according to a new Table 1.
Change the code names for the cultivars as C1 to C13 might be mistaken for the number of carbon atoms in complex carbon-based molecules. The authors should use different initials to represent the genera and species, like 'Chmor' or 'Cmo'.
Response: ‘Cm’was used to avoid the confusion.
Figure legends: Please do not list all the names for C1-C13. The authors could direct the reader to refer to Table 1, as was done for the compounds in Fig. 2 legend.
Response: In the Figure legends, C1-C13, now Cm1-Cm13, was referred to Table 1 as suggested.
Figure 2: The frequency ranges from 1 - 13, which should be easy to show instead of using percentage on Y-axis.
Response: The number of cultivars (ranging from 1-13) on Y-axis was used to replace percentage as suggested.
Figure 3B: The use of bracket lines to show groupings in the HCA does not match with the PCA. Please fix the figure.
Response: The bracket was adjusted.
Lines 144-185: It is very descriptive. It would be really interesting if the authors could show that the amounts of some of these apolar compounds are statistically/significantly different between the three groups based on PCA/HCA.
Response: We did some additional analysis and added the following description: “By comparing the PCA/HCA results based on terpenoid contents and antifungal activities, the antifungal effects of several cultivars (including ‘Da yang ju’(Cm 5), ‘Huang xiang li’(Cm 6), ‘Huang ju’(Cm 8), ‘She yang hong xin ju’(Cm 12)) could be associated with several terpenoids, including zingiberene, β-sesquiphellandrene, α-curcumene, (E)-β-farnesene and α-farnesene. To establish causal correlation between these individual terpenes and antifungal activities will be an interesting future research.”
What is the basis of these differences? Is it genetic, population structure? The authors mention this in one line in the conclusions. Please expand on it a bit more.
Response: this discussion point was expanded.
The fungal isolates tested in this study, where were those obtained from? Were those isolated from Chrysanthemum?
Response:all information is now provided as suggested.
Figure 4D: What is CK? Please add to foot-note. In methods, only two artificial media, PDA and Malt extract, are mentioned.
Response:‘CK’was replaced by ‘Control’. Control refers to organic solvent ethyl acetate, which was used for leaf extraction. The assays with ethyl acetate served a negative control. This is now indicated in both Method and Figure legend.
Lines 213-215: The name of the cultivar mentioned in the text and that shown in Fig. 4D does not match.
Response:The names in the main text and figures are now consistent.
Lines 293-305: How many plates were used per replicate?
Response:Number of plates for the each replicate (10) was added.
In the HCA figures, Fig. 3B, Fig. 5B: What is the meaning of the dotted line?
Response:The dotted line was unnecessary and was deleted.
Minor comments:
----- ---------
Line 102: typo 'apola'
Response:‘Apola’was corrected as ‘apolar’.
Make Table 2 supplementary
Response:Table 2 was changed to be Supplementary Table 1.
Reviewer 4 Report
Since there is few new findings from the following basis, the referee would not agree with publication of this manuscript.
It is well known and well studied that the extracts from many Asteraceae genus plants have anti-fungal activities. The authors were analysed the anti-fungal effects and the terpenoid content of the extract from 13 cultivated plants using GCMS as well as the PCA and HCA analysis. However, these terpenoids are common components in the plant generally, and there are many reports on thiry antifungal activity.Author Response
Since there is few new findings from the following basis, the referee would not agree with publication of this manuscript.
It is well known and well studied that the extracts from many Asteraceae genus plants have anti-fungal activities. The authors were analysed the anti-fungal effects and the terpenoid content of the extract from 13 cultivated plants using GCMS as well as the PCA and HCA analysis. However, these terpenoids are common components in the plant generally, and there are many reports on thiry antifungal activity.
Response: We respectively disagree with the evaluation of the reviewer.
Round 2
Reviewer 1 Report
No more comments
Author Response
Thanks for the positive comments!
Reviewer 4 Report
As for the anti-fungal activities, any suitable positive agents under the same condition should be performed and described them. Preferably, if the anti-fungal activities of these extracts obtained from C. morifolium in this manuscript can be compared to some naturally occurring extracts that are well recognized for their anti-fungal activity, the authors might be able to appeal the usefulness.
Author Response
Response to reviewer 4:
We thank the comment. If this the objective of this study was to commercialize chrysanthemum leaf extracts as antifungal agent, then the comment is most appropriate. However, it is out of the scope of this study. Nonetheless, we added the following sentence to acknowledge its importance for future research.
“To this end, it will be important to compare the antifungal activities of Chrysanthemum leaf extracts with those of comparable commercial products.”